# Youth friendly sexual and reproductive health service utilization among high and preparatory school students in Debre Tabor town, Northwest Ethiopia: A cross sectional study

**Amare Simegn[1], Telake Azale[2], Abebaw Addis[2], Mulugeta Dile[1], Yitayal Ayalew[1], Biniam Minuye****[1]***

**1** College of Health Sciences, Debre Tabor University, Debre Tabor, Ethiopia, **2** College of Health Sciences, University of Gondar, Gondar, Ethiopia

* biniamminuye@yahoo.com

## Abstract

### Background

Youth continue to fall victim to sexual and reproductive health problems. Despite, reproductive health needs of youth had been supported by different organizations, utilization of those services is low. All efforts have not been felt across the Ethiopian learning institutions as is evidenced by persistent reproductive health problems. Therefore, this study aimed to assess the magnitude of youth friendly sexual and reproductive health service utilization and associated factors among high and preparatory school youths in Debre Tabor town, Northwest Ethiopia.

### Methods

An institution based cross- sectional study was conducted from March 1 to 28, 2016. The data were collected using a pre-tested and structured self-administered questionnaire. Multistage cluster sampling method was used to select the study participants. The data were entered into Epi-data version 4.2.0.0 and analyzed using SPSS version 20. Binary logistics regression was used for analysis. Odds ratio along with 95%CI was estimated to measure the strength of the association. Level of statistical significance was declared at p value ≤0.05.

### Results

Overall utilization of reproductive health service was 28.8%. Being male (AOR = 1.54, 95% CI: 1.05, 2.25), prior discussion on reproductive health issues (AOR = 6.33, 95% CI: 4.22, 9.51), and previous sexual intercourse within the past one year (AOR = 1.95, 95% CI: 1.10, 3.44) were significantly associated with youth friendly health service utilization.

**Data Availability Statement:** All relevant data are within the manuscript.

**Funding:** The author(s) received no specific funding for this work.

**Competing interests:** The authors have declared that no competing interests exist.

**Abbreviations:** ANC, Antenatal Care; AOR, Adjusted Odds Ratio; CI, Confidence Interval; FMOH, Federal Ministry of Health; FP, Family Planning; HIV, Human Immune Virus; NGO, Non-Governmental Organizations; OR, Odds Ratio; RH, Reproductive Health; SPSS, Statistical package for social sciences; SRH, Sexual and Reproductive Health; VCT, Voluntary Counselling and Testing; WHO, World Health Organization Declarations.

## Conclusions

Youth friendly health service utilization among high school and preparatory students in Debre Tabor town was low. Ensuring gender empowerment and advocating sexual and reproductive service discussion among themselves and with others might be important in improving reproductive health utilization and health. Future researcher should address segment of population who does not enter school.

## Background

Reproductive health is "a state of complete physical, mental and social well-being and not merely the absence of disease or infirmity, in all matters relating to the reproductive system." it is critical during adolescence and adulthood, and affects the health of the next generation [1]. Every year an estimated 1.7 million youths lose their lives prematurely related to reproductive health problems [2]. Due to these reasons, adolescent's reproductive health (RH) is becoming ever more important component of global health.

Youth are characterized by significant physiological, psychological and social changes. This shares about 20% of the world's population [3, 4]. More than one third of the Ethiopian population is aged between 15–24, mostly vulnerable to a range of sexual and reproductive health problems [5].

Focusing on adolescent reproductive health is both a challenge and an opportunity for health care providers. Adolescence generally is a healthy period of life, many youths are less informed on reproductive health services, less experienced, and less comfortable accessing health services for reproductive issues than adults [6–8]. Limited access and low utilization to targeted reproductive health care and services for young people contributes and exacerbates many of the RH problems [9]. These problems could be an early childbearing, early sexual debut, sexually transmitted infections (STIs) including HIV/AIDS, unmet need for family planning, early marriage, and abortion [9]. Over a quarter of all pregnant youth and adolescents feel that their pregnancies are mistimed, reflecting this population's limited access to family planning (FP) and other RH services [10]. The reasons for lower reproductive health service (RHS) utilization may include feelings of discomfort, fear of being seen by parents and embarrassment while seeking reproductive health services [11].

Efforts have been made to address different reproductive health problems at different level. The Ethiopian Government, along with a number of international non-governmental and local organizations has been supporting activities to increase access to sexual and reproductive health (RH) services for young people. This includes the scaling-up and institutionalization of Youth Friendly Services (YFS) through intensive capacity building at all levels of the health system [12, 13]. However, the effects of all these efforts have not been well understood across the Ethiopian high schools and preparatory schools as is evidenced by persistent reproductive health problems and challenges to the youth [14–16].

Utilization of reproductive health services is the key for the improvement in the quality of life of the youth. Behaviors formed and choices made during adolescence period. Despite different studies conducted in Ethiopia, existing data didn't fully assess the factors which affect service utilization. Therefore, this study assessed the magnitude and associated factors for Sexual and reproductive health services utilization among students in Debre Tabor town.

## Materials and methods

### Study design, period, setting, and population

An institution based cross sectional study was employed in Debre Tabor town, Northwest Ethiopia, from March 13 to 18, 2016. Debre Tabor town is located in South Gondar Administrative Zone of the Amhara regional, state, at a distance of 667 kilometers away from Addis Ababa, the capital city of Ethiopia, 100 kilometers southeast of Gondar and 50 kilometers east of Lake Tana [17]. There are five privately owned kindergartens, 12 government and private primary schools, four government senior secondary schools (9–10), two preparatory schools, one Teachers vocational educational training, two public and one private college, and one University. In the town, there were 4152 (2242 female and 1910 male) high schools and 2955 (1459 male and 1496 female) preparatory students.

All youths between the age range of 15 to 24 years who were attending their education in secondary and preparatory schools during the study period were the source population. Night and extension students were excluded. A total of 696 students participated in the study. Samples were drawn from six school with proportional allocation method. Finally, the simple random sampling technique was applied to get the required number of eligible students.

### Sample size calculation

The sample size was determined by using Epi info 7 statistical software by single population proportion based on following Assumption; Estimated proportion (p) taken from previous study done at hadiya zone on the utilization of reproductive health services among youth (29.4%), Margin of error d = 5%, Confidence level of 95%. The final sample size was 696.

### Sampling technique

Multi stage cluster sampling method was used to select the study participants. There were 4 high schools and 2 preparatory schools in Debre Tabor town. The calculated Sample size was proportionally allocated for selected schools. Then, 24 sections (6 sections for each high school) from a total of 81 high school sections were randomly selected. Similarly, 19 sections were randomly selected from total of 64 preparatory sections. The students from these eligible sections were randomly selected using the list of student's ID number as a sampling frame. Finally, the data was collected from 400 high school students and 290 preparatory students. (Fig 1).

### Data collection

Data were collected by using structured close ended self-administered questionnaire. A total of 37 questions which consists of socio demographic, family characteristics, respondent's awareness and sources of information about the services, health system factors and youth preferences for time and health care workers related factors. Data were collected by six nurses and supervised by three public health professionals. The questionnaire was prepared in English and translated to "Amharic ".'Forward and backward translation' was done. Completeness of each recording format was checked before collecting the data.

### Variables

**Dependent variable.**    Youth friendly health service utilization.

**Independent variable.**    Socio demographic, family characteristics, respondent's awareness and sources of information about the services, and youth preferences for time and health care workers related factors.

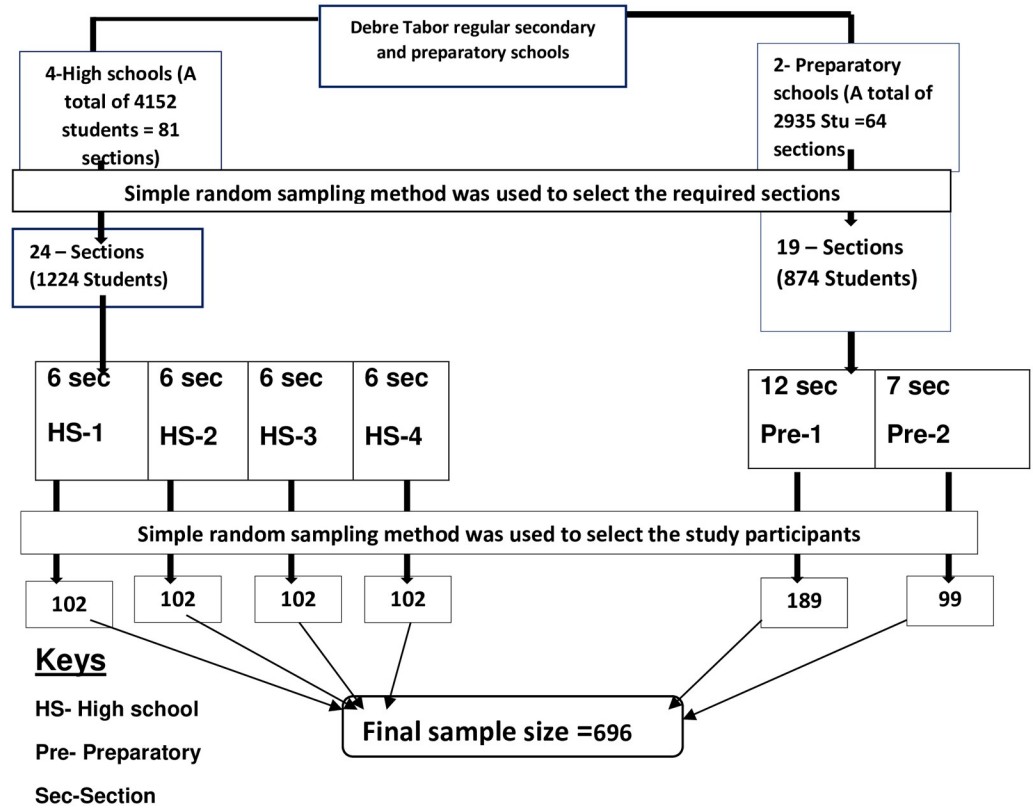

**Fig 1. Schematic presentation of sampling procedures.**

## Operational definitions

- Reproductive health service utilization was assessed on the basis of youth practice of utilizing reproductive health services in the past one year, which includes family planning, sexually transmitted infections, diagnosis and treatment on sexual and reproductive related problems, voluntary counselling and testing, antenatal care, abortion care, post abortion care, condom use and general health information/counselling. Thus, using at least one component of the stated reproductive health services within the past one year was considered as the SRH service is utilized by the students.

- Sexual activity: Refers to the involvement in a penetrative heterosexual intercourse within the last 12 months.

Preparatory students: Those students who are in grades 11 and12.
   High school: A school, which entails grades 9 and 10.

- Awareness about SRH Services: Those knowing the availability of one or more RH service were reported as having awareness about the services.

- Youth preferences: Preference of youth in relation to health care workers and time for using SRH services.

## Data quality control

Pretest was done on 5% of the sample size. Training was given for two consecutive days on how to administer questionnaires, handling ethical issues and maintaining confidentiality and privacy. Completeness of each data collection tools had been checked by the principal investigator and the supervisors in a daily base. Double data entry was done by two data clerks and consistency of the entered data was cross-checked.

## Data processing and analysis

Data were entered, coded, cleaned and checked by EpiData statistical software version 4.2.0.0 and analysis was done by using SPSS Version 20 statistical software. Binary logistic regression was used for analysis. Bivariable analysis was done and all variables that were found to be significant at p-value <0.25 were entered into the multivariable logistic regression model. Independent variables that were significant at p-value <0.05 in the multiple logistic regression models was considered as statistically significant. Finally, the data were presented in texts, figures and tables.

## Ethical considerations

Ethical clearance was obtained from Gondar University, College of Medical and Health Sciences, Institutional Health Research Ethics Review Committee (IHRERC)Then Official letter had been written to each school for permission and support. Permission letter was obtained from the selected schools. The purpose of the study was explained to the study participants, informed written consent was taken. For children less than 18 years assent (written consent) were secured from their guardians/parents and confidentiality of the information was ensured. Data collected was used only for the study purpose.

# Results

## Socio demographic characteristics of the respondents

In the study a total of 690 students were included in the study, making a response rate of 99.1%. More than half, 376 (54.5%) were females. More than three fourth, 549 (79.6%) were within the age range of 15–19 years. (Table 1).

## Respondent's family characteristics, living arrangement and sources of information

Four hundred and eighty-two (70.2%) of the respondent's mothers had no formal education. More than half, 370 (53.6%) of the respondent's fathers were farmers. One hundred eighty two, (26.2%) of the respondents were living alone. Two third, (73.2%) of the respondents had sources of information from radio and television (Table 2).

## Respondent's awareness about reproductive health services

Six hundred and twenty (90%) of the respondents had awareness at least about one service, and 542 (78.6%) had information about the presence of a RH service facility in the town. However, only 30.9% of them had awareness about counselling services.

　　Youths' preferences to time for seeking reproductive health services, to type of health care providers, service fee and characteristics of health facilities.

　　Two hundred and ninety-seven (43%) of the students preferred weekend time as the most favorable service hour and 286 (41.4%) preferred youth with the same sex as a health care

**Table 1. Socio demographic characteristics students, Debre Tabor, Ethiopia, 2016.**

| Variables | Frequency | Percentage |
|---|---|---|
| Sex | | |
| Male | 314 | 45.5 |
| Female | 376 | 54.5 |
| Age | | |
| 15–19 | 549 | 79.6 |
| 20–24 | 141 | 20.4 |
| Ethnicity | | |
| Amhara | 684 | 99.1 |
| Others | 6 | 0.9 |
| Religion | | |
| Orthodox | 672 | 97.4 |
| Muslim | 13 | 1.9 |
| Others | 5 | 0.7 |
| Educational level | | |
| Grade- 9 | 115 | 16.7 |
| Grade-10 | 285 | 41.3 |
| Grade-11 | 144 | 20.9 |
| Grade-12 | 146 | 21.2 |
| Residence | | |
| Rural | 372 | 53.9 |
| Urban | 318 | 46.1 |

provider. Three hundred and seventy-nine (54.9%) of the respondents described the health care providers as good and welcoming. However, 90(13.0%) of youth were mistreated by health care providers.

Two hundred ninety four, 42.6% of students responded that the health care facility found in their respective residence was not suitable to use the services due to the mistreating by health care providers, long distance and unfavorable service hours (Table 3).

## Reproductive health services utilization

Overall reproductive health utilization was 28.8%) [95% CI: 25.5–32.5]. Family planning, voluntary counselling and testing for HIV, STI diagnosis and treatment and general counselling services were utilized by 14.9%, 10.5%, 10.3% and 7.9% youths respectively (Fig 2).

## Factors associated with reproductive health service utilization

In bivariate analysis, participant's sex, residence, awareness of RH facility, awareness of RH service, discussion on RH services, ever had a sexual partner and penetrative sexual intercourse within the last one year were identified as significant predictors of reproductive health service utilization.

However, multivariable logistic regression showed three variables, i.e being male, penetrative sexual intercourse within the last one year and adolescents who had a discussion about reproductive health issues were associated with reproductive health service utilization.

Males were about 1.5 times more likely to utilize reproductive health services as compared to females (AOR = 1.54, 95% CI: 1.05, 2.25). Adolescents who had a discussion about reproductive health issues were about 6.33 times more likely to use reproductive health services

**Table 2. Family characteristics, living arrangements and communication means of students in Debre Tabor town, Northwest Ethiopia, 2016.**

| Family characteristics | Frequency | Percentage |
|---|---|---|
| Mother's educational status | | |
| Unable to write and read | 215 | 31.2 |
| Able to write and read | 267 | 38.7 |
| Complete primary and secondary education | 148 | 21.4 |
| Diploma and above | 60 | 8.7 |
| Father's educational status | | |
| Unable to write and read | 111 | 16.1 |
| Able to write and read | 339 | 49.1 |
| Complete primary and secondary education | 131 | 19.0 |
| Diploma and above | 109 | 15.8 |
| Mother's occupational status | | |
| House wife | 520 | 75.4 |
| M /trade vendor | 66 | 9.6 |
| Civil servant | 75 | 10.9 |
| Others* | 29 | 4.1 |
| Fathers occupational status | | |
| Farmer | 370 | 53.6 |
| Market trade vendor | 130 | 18.8 |
| Civil servant | 129 | 18.7 |
| NGO | 31 | 4.5 |
| Others** | 30 | 4.3 |
| Family alive | | |
| Mother | 69 | 10.0 |
| Father | 23 | 3.3 |
| Both alive | 596 | 86.4 |
| Both died | 2 | 0.3 |
| Respondent's living arrangement | | |
| With both parents | 364 | 52.8 |
| Privately rented | 182 | 26.2 |
| Mother only | 59 | 8.6 |
| Father only | 15 | 2.2 |
| Living alone | 34 | 4.9 |
| With relatives | 51 | 7.4 |
| Others*** | 19 | 2.8 |
| Communication means in the respondent's house | | |
| Radio | 213 | 30.9 |
| Television | 292 | 42.3 |
| No communication means | 174 | 25.2 |
| Others**** | 11 | 1.6 |

Note:

* NGO, daily laborer,

**daily laborers, car drivers, carpenters,

*** with sister, brother, sexual partner;

**** cell phone, newspaper, pamphlets.

**Table 3. Youths preferences to time, to health care providers, service fee and characteristics of health facility in Debre Tabor town, Northwest Ethiopia, 2016.**

| Type of choice | Frequency | Percent |
|---|---|---|
| Time preference of youth | | |
| 1. Any time | 255 | 37.0 |
| 2. Before school/morning | 73 | 10.4 |
| 3. After school/noon | 65 | 9.3 |
| 4. Weekend | 297 | 43.3 |
| Person preferences of youths | | |
| 1. Youth with the same sex | 286 | 41.4 |
| 2. Youth with any sex | 154 | 22.3 |
| 3. Adult with the same sex | 53 | 7.7 |
| 4. Any health care provider | 197 | 28.6 |
| Service fee | | |
| 1. As usual rate | 139 | 20.1 |
| 2. With discount for the students | 101 | 14.6 |
| 3. Free of charge | 450 | 65.3 |
| General characteristics of the HF in the town | | |
| Comfortable | | |
| 1. Yes | 396 | 57.4 |
| 2. No | 294 | 42.6 |

than those who had not discussed with health care workers, family, teachers, peers, and sexual partners(AOR = 6.33, 95% CI: 4.22, 9.51). In addition, adolescents who had penetrative heterosexual intercourse within the past one year were also nearly 2 times more likely to use the services than their counterparts(AOR = 1.95, 95% CI: 1.10, 3.44) (Table 4).

## Discussion

The study revealed that the overall utilization of reproductive health services by high school and preparatory school students was found to be 28.8%, which was consistent with studies conducted in Badewacho woreda, southwest Ethiopia (29.4%) [18] and in Bahir Dar (32%) [19]. However, the findings of the present study is lower than the studies carried out in Dejen district, east Gojjam, Ethiopia,(45%) [20], in Jimma (34.7%) [21], and in Harar, Ethiopia, (63.8%) [22]. The possible explanations for the difference of the study in Harar could be justified as a higher proportion of married respondents (16.3%) in Harar may result in a higher proportion of service utilization. Furthermore, lower as compared to the studies done in Botswana [23] and in England [24]. This might be due to differences in the availability and accessibility of youth friendly health facilities, youth centers, educational status, socioeconomic status, type of residence, transportation and culture.

In this study sex, partner discussion about reproductive health issues and sexual intercourse were significantly associated RH service utilization. The study revealed that males were more likely to utilize reproductive health services as compared to females. This finding is in conformity with the study conducted in Nepal [25] and in Addis Ababa [9]. Nonetheless, it is different from the finding in Badewacho woreda, Hadiya zone, Ethiopia [18]. This might be due to cultural influences in the study area in which females was less empowered [26, 27] to go to health facilities for reproductive health services. It might also be due to the fact that the proportion of male and female participants in our study was different from Badewacho woreda's (Male: Female, 50.9:49.1).

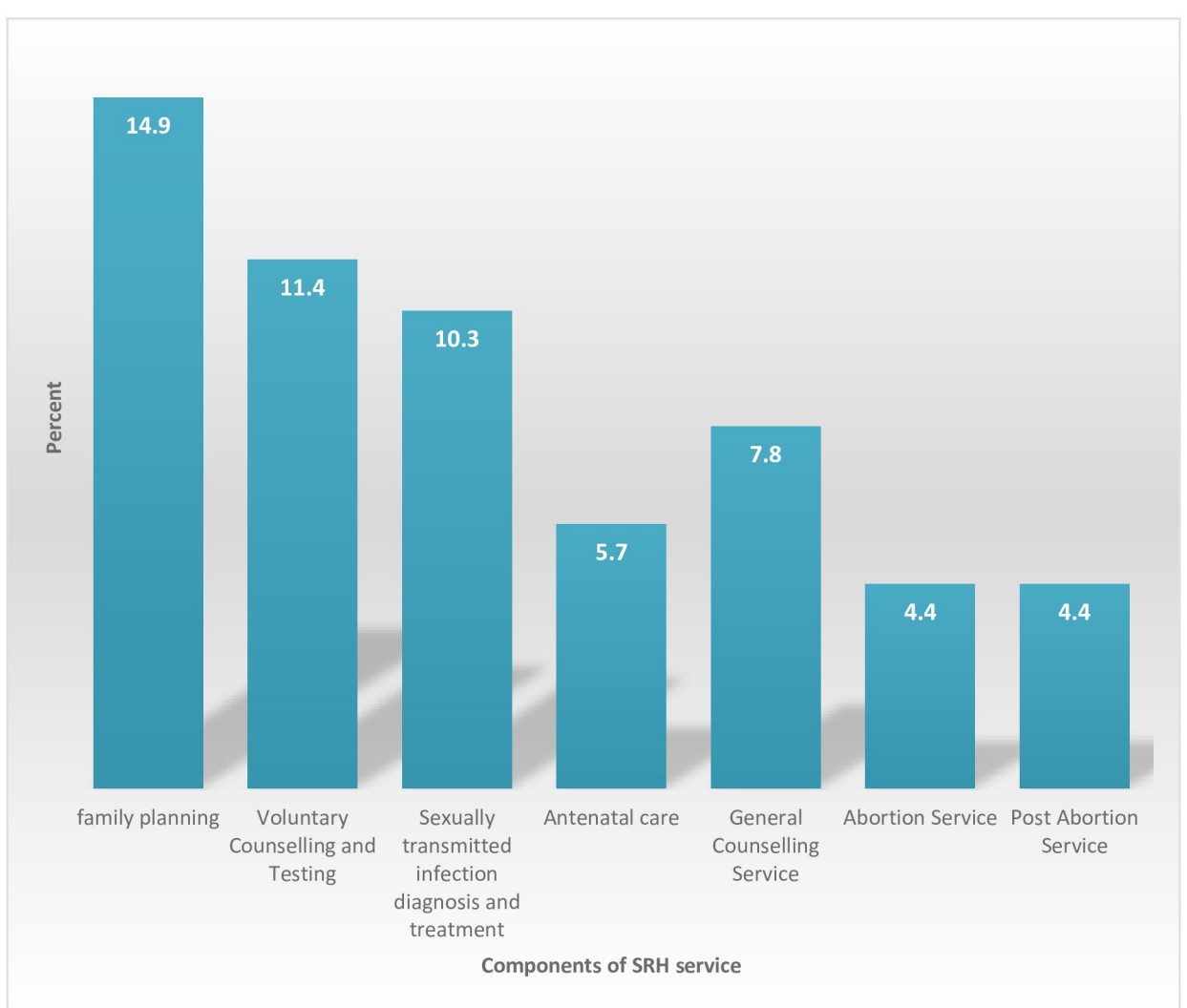

**Fig 2. Components of reproductive health services mentioned by high school and preparatory school students, Debre Tabor, Ethiopia, 2016.**

This study also indicated that youth who had discussions about RH services with health workers, family, teachers, peers, and sexual partners were more likely to utilize the services than their counterparts. The finding of this study is consistent with the findings in Gondar [28] and in Madawalabo [29]. This might be due to prior information on RH service increase their utilization, which would in turn motivate them to use the service. It might be also due to the possibility that youth who use services are also more likely to discuss services.

Students who had sexual intercourse within the past one year were more likely to utilize RH service than abstainers. This finding is similar with the studies carried out in Gondar [28] and in West Badewacho woreda, Ethiopia [18]. This might be due to the fact that many of reproductive health service components (contraception) might be used when youths have perceived reproductive health risks related to sexual intercourse.

This study also tried to highlight that being too young to go for the services and inconvenient service hour were the two main reasons for them not to use sexual and reproductive health services for the students. This is similar with the study done in Bahir Dar, in Addis Ababa and in Hara [30–32]. This might probably be due to the reason that students spent their

**Table 4. Factors associated with reproductive health service utilization among high school and preparatory students in Debre Tabor town, Northwest Ethiopia, 2016.**

| Variables | RH services utilization | | Crude OR (95% CI) | AOR (95% CI) |
|---|---|---|---|---|
| | **Yes** | **No** | | |
| Sex | | | | |
| Male | 106 | 208 | 1.56(1.11–2.16) | 1.54(1.05–2.25) * |
| Female | 93 | 283 | 1.00 | 1.00 |
| Age | | | | |
| 15–19 | 157 | 392 | 0.94(0.63–1.42) | 0.94(0.59–1.52) |
| 20–24 | 42 | 99 | 1.00 | 1.00 |
| Residence | | | | |
| Rural | 122 | 250 | 1.53(1.09–2.14) | 1.40(0.91–2.08) |
| Urban | 77 | 241 | 1.00 | 1.00 |
| Participant's educational level | | | | |
| High school | 125 | 275 | 1.33(0.95–1.86) | 1.30(0.93–2.03) |
| Preparatory | 74 | 216 | 1.00 | 1.00 |
| Respondent's awareness on RH facility | | | | |
| Yes | 173 | 369 | 2.20(1.39–3.49) | 1.70 (0.99–2.92) |
| No | 26 | 148 | 1.00 | 1.00 |
| Respondent's awareness on RH services | | | | |
| Yes | 187 | 433 | 2.09(1.20–3.98) | 0.87(0.41–1.84) |
| No | 12 | 58 | 1.00 | 1.00 |
| Respondent's discussion of RH service | | | | |
| Yes | 159 | 173 | 7.30(4.93–10.80) | 6.33(4.22–9.51) ** |
| No | 40 | 318 | 1.00 | 1.00 |
| Ever had a sexual partner | | | | |
| Yes | 66 | 97 | 2.02(1.40–2.92) | 1.36(0.84–2.19) |
| No | 133 | 394 | 1.00 | 1.00 |
| Sexual intercourse within the last one year. | | | | |
| Yes | 46 | 46 | 2.91(1.86–4.55) | 1.95(1.10–3.44) * |
| No | 153 | 445 | 1.00 | 1.00 |

Note:

* significant at a p-value of $< 0.05$,

** significant at a p-value of $< 0.01$.

time at school during the regular health institutions' working hours and the institution may not be possibly functional in the weekend at which the students are relatively free. Besides, since the society declares the students as they are too young to go to the health institution due to some cultural influences and visiting health institutions for particular SRH services might be thought as shameful.

Furthermore, it is found that lack of knowledge on the advantages of the services, long queue, inconvenient location of health facilities, lack of money and the mistreating behavior of health care providers were found to be important barriers for youths to utilize sexual and reproductive health services. This is in line with the finding in Malawi and Kenya [33, 34]. This might be explained by the fact that most of the students may have little or no awareness about most of the sexual and reproductive health services are being offered free of charge. Besides, the students may also perceive that long queue may let them for unnecessary exposure to the peoples who might be around the health institution and this exposure might leave

frustration related to their privacy and confidentiality. Hence, the situation might be exacerbated if they think that the location of the health institution is inconvenient for them. Therefore, educating female students to go to health facilities for SRH services, creating awareness on the preventive aspects of SRH problems and advocating SRH service discussion among themselves and with others are important. Since the study is cross-sectional it does not show cause effect relationship. Those segment of population who did not enter school were missed. Factors from the service providers' perspective, structural barriers as well as the supply was missed.

## Conclusions

Reproductive health service utilization among high school and preparatory students in Debre Tabor was low. Youth who are male, prior discussion on RH services to health workers, family, teachers, peers, and sexual partners and having penetrative heterosexual intercourse within the past one year were found to have significantly associated with reproductive health service utilization. So future research should be done on effect of poor utilization of reproductive health services on health outcome by including variables such as service providers' perspective and structural barriers.

## Supporting information

**S1 Dataset.**
(RAR)

## Acknowledgments

The authors acknowledged students who participate in the study and data collectors.

## Author Contributions

**Formal analysis:** Amare Simegn.

**Investigation:** Amare Simegn.

**Methodology:** Amare Simegn, Telake Azale, Abebaw Addis, Yitayal Ayalew.

**Supervision:** Telake Azale, Abebaw Addis.

**Writing – original draft:** Amare Simegn.

**Writing – review & editing:** Amare Simegn, Mulugeta Dile, Yitayal Ayalew, Biniam Minuye.

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
