## [Decision Letter · Decision Letter 0]

19 Mar 2020

PONE-D-19-33641

Youth friendly health service utilization among high and preparatory school students in Debre Tabor Town, Northwest Ethiopia.

PLOS ONE

Dear Mr. Minuye,

Thank you for submitting your manuscript to PLOS ONE. After careful consideration, we feel that it has merit but does not fully meet PLOS ONE’s publication criteria as it currently stands. Therefore, we invite you to submit a revised version of the manuscript that addresses the points raised during the review process.

We would appreciate receiving your revised manuscript by May 03 2020 11:59PM. To enhance the reproducibility of your results, we recommend that if applicable you deposit your laboratory protocols in protocols.io, where a protocol can be assigned its own identifier (DOI) such that it can be cited independently in the future. For instructions see: http://journals.plos.org/plosone/s/submission-guidelines#loc-laboratory-protocols

We look forward to receiving your revised manuscript.

Kind regards,

Julie Maslowsky, PhD

Academic Editor

PLOS ONE

Journal Requirements:

2. Please include additional information regarding the survey or questionnaire used in the study and ensure that you have provided sufficient details that others could replicate the analyses. For instance, if you developed a questionnaire as part of this study and it is not under a copyright more restrictive than CC-BY, please include a copy, in both the original language and English, as Supporting Information.  If the original language is written in non-Latin characters, for example Amharic, Chinese, or Korean, please use a file format that ensures these characters are visible.

3. Please state whether you validated the questionnaire prior to testing on study participants. Please provide details regarding the validation group within the methods section.

4. You indicated that you had ethical approval for your study. In your Methods section, please ensure you have also stated whether you obtained consent from parents or guardians of the minors included in the study or whether the research ethics committee or IRB specifically waived the need for their consent.

Additional Editor Comments:

I and four reviewers agree that although your study shows some promise for publication, substantial revisions are needed. Please respond thoroughly to the comments and suggestions provided by the reviewers.

Reviewers' comments:

Reviewer's Responses to Questions

**Comments to the Author**

1. Is the manuscript technically sound, and do the data support the conclusions?

Reviewer #1: Yes

Reviewer #2: Yes

Reviewer #3: Yes

Reviewer #4: Partly

2. Has the statistical analysis been performed appropriately and rigorously? 

Reviewer #1: No

Reviewer #2: Yes

Reviewer #3: Yes

Reviewer #4: No

3. Have the authors made all data underlying the findings in their manuscript fully available?

Reviewer #1: Yes

Reviewer #2: Yes

Reviewer #3: Yes

Reviewer #4: No

4. Is the manuscript presented in an intelligible fashion and written in standard English?

Reviewer #1: No

Reviewer #2: Yes

Reviewer #3: No

Reviewer #4: Yes

5. Review Comments to the Author

Reviewer #1: Summary of the article

The manuscript "Youth friendly health service utilization among high and preparatory school students in Debre Tabor Town, Northwest Ethiopia" tries to assess utilization and factors affecting utilization among youths. The study may have minimal contribution to the context of the study area if the following major and minor comments are addressed by the:

1. General comment

• First of all, the issue is overlooked in various parts of the country. Some of these includes Jimma (South West Ethiopia), Harar (East ethiopia) , Anchar district (East ethiopia ) , Ethiopia (Meta analysis ), Badwacho (Southern Ethiopia), Wolaita ( Southern Ethiopia), Hossana ( Southern Ethiopia), Asela ( South East Ethiopia), Bahir Dar ( North Ethioia ), Tigray ( North Ethiopia), Debre Birhan ( North Ethiopia), Nekemte ( West Ethiopia), Awabel district ( North west Ethiopia), Metekel Zone ( North West Ethiopia), Goba town ( South East Ethiopia), Dejene district ( North Ethiopia), Debre Markos (North Ethiopia), Woreta Town ( North West Ethiopia), Kachabira District ( Southern Ethiopia), Meachkal Distric ( North Wets Ethiopia), and Mecha district ( North west Ethiopia). If we commutate all findings, the utilization of “youth friendly sexual and reproductive health service” is very low. So what ? Does this study come with a new perspective ? Yes, we know the contexts particularly for these subgroup populations are different in different areas due to individual level factors, structural factors, sociodemographic factors, and so on. Majority of the studies including the current study are done in towns and all are from youth/adolescent perspectives of the services.

• The justification “the status of these segments of population in using sexual and reproductive health services is not clearly known in the country” in the abstract section is not palatable and therefore, need another peculiar justification.

• There are issues related to the journal’s guideline like reference writing consistency

2. Abstract

• See the track change comments

3. Methods section

• See the track change comments

4. Results Section

• See the track change comments

5. Reference and other issues

• See the track change comments

Reviewer #2: Comments

This is important topic which highlighted issues among youth.

I would like to provide my comments to improve this paper further.

Background

1. Generally you have mentioned the important point regarding issue in this study, however literature seems very brief. I would suggest to add on more

2. Paragraph 3… ‘They lack RH information, knowledge and access………service for RH’ is not clear. Maybe you could restructure the sentences

3. The acronym used in highlighting the issues not consistent as you used for SRH, RH and ARH. My suggestion is used consistency term and acronym that best applied to this study. Using full terminology rather than acronym is much better.

4. In Paragraph 6, acronym for YFS means what?

Methodology

1. How many population in this town and may also include how big is the town.

2. Please elaborate detail on sample size calculation; how you get 696 students participated in this study. How you did simple random sampling? How many school involved?

3. Study instrument is not clear and need further elaboration, eg what questionnaire you used, how many questions, how it is developed and does it is validated

4. In writing the variables I suggest to use proper sentences. Also maybe it is very nice to put point or numbers in writing the operational definition or maybe just highlighted the variable /operational definition related to your result and discussion rather put it all as listed

5. In ethical consideration, since your sample /participants are between age 15 to 24 years old, the ethical address should properly highlighted. Maybe you should mention at what age respondent can give consent and for those under children age (less than 18 years) how consent form was taken and should also mention others ethical issues as this topic or question that you asked give impact to subject.

Result & Discussion

1. Is well presented however should report detail on response rate in this study? Maybe report the reason why they rejected to participate?

2. In the discussion , paragraph 2 …(M:F, 50.9:49.1)….what does it means? Is it references?

However, maybe this study only represented youth who are in the education system since your population is not cover among those who did not enter school. This should be discuss in the limitation and maybe suggestion to improve in research.

Conclusion & Limitation -

Should explain detail

Reviewer #3: Overall, this is a clear, concise and well-written manuscript. The topic is timely and relevant. Below are my comments:

Background

Page 1, line 1: Grammar error - "Reproductive health is 'a'..."

Page 1, line 1-2: Definition of ‘reproductive health’ – this is a direct quote. Please include quotation marks.

Page 1, line 4: "Every year an estimated 1.7 million youths lose their lives prematurely". Were you referring SRH problems? Please specify.

Page 1, para 2: "Individuals in the age group of 15-24 years are characterized by significant physiological, psychological and social changes and making up about 20% of the world’s population, 85% lived in developing countries" - This sentence is too long and confusing. Suggest to rephrase the sentence.

Page 1, para 3: adolescent reproductive health (RH) - please be consistent with the abbreviation used. ARH or RH?

Methods

Page 2, Study period, Setting, and Population, para 1 - The study duration is not the same as stated in the abstract.

Why Debre Tabor town was chosen as the study setting? Please justify.

Page 3, para 1 – “There are five privately owned kindergartens, 12 government and private primary schools, four government senior secondary schools (9-10), two preparatory schools, one Teachers vocational educational training, two public and one private college, and one University”- Are all these schools and colleges located in Debre Tabor town? Please be clear.

Page 3, para 1: "In the town, there were 4152 (2242 female and 1910 male) high schools and 2955 (1459 male and 1496 female) preparatory students" - Are high schools the same as secondary schools? This needs to be clarified from the start.

Page 3, para 2 - "Night and extension students were excluded" - why were they excluded?

Page 3, para 2 - "A total of 696 students were participated..." - Grammar error

Page 3, para 2 - "Samples were drown..." - drawn

How did the author do simple random sampling? Was random table or computer used to generate the samples?

Why are there two subtitles? - Data Collection Methods & Data Collection Procedure. Suggest 'Data Collection'.

Page 3, last paragraph - "Data were collected by using structured close ended self-administered questionnaire. Data were collected by six nurses during working hours’ and supervised by three public..." - 'data were collected' used in two consecutive sentences.

Page 3, last paragraph: "Data were collected by six nurses during working hours..." - is it necessary to state the 'working hours'?

Page 3, last paragraph: The questioner was prepared in English and translated in “Amharic” for data collection and retranslated into “English” - spelling error questionnaire and you may want to use the term 'forward and backward translation' when describing the translation process.

Page 4: Dependent variable - Youth friendly health service utilization - there is no description of YFHSU. How was it measured?

Family characteristics - how was it measured?

Is the term definition for 'Reproductive health service utilization' the same as 'Youth friendly health service utilization'? Please be consistent.

Page 4: Were the variables defined under 'Operational Definitions' adopted from a validated questionnaire or did the author developed their own questionnaire? This needs to be clearly stated in the manuscript.

Page 4: Youth-communication/discussion: “Youth report having talked to anyone else about one or more SRH services were categorized as having a communication/discussion about the services” - please specify the people whom they had conversations with e.g. family, teachers etc.

Page 5, Youth preferences: 'Preference of youth in relation to health care workers' and 'time focusing SRH services' - were these two sub-variables separated in the questionnaire?

How was sample size calculated? This needs to be described.

Ethical considerations - was parental consent obtained? In some countries e.g. Malaysia, those less than 18 years old are considered minors and parental consent is required.

Ethical considerations - The purpose of the study was explained to the study participants, informed written consent and assent were secured and confidentiality of the information was censured - why confidentiality was ‘censured’? Did you mean to say assured?

Results

Page 6, 'sources of information' - please specify. Was it sources of info on SRH?

Page 7, RHSU: A total of 690 school youths were participated. – grammar error and avoid repetition.

Discussion

Page 8: “The possible explanations for the difference of the study in Harar could be justified as a higher proportion of married respondents (16.3%) in Harar may result in a higher proportion of service utilization”. - To include in-text citation.

Page 8-9: “This might be due to differences in the availability and accessibility of youth friendly health facilities, youth centers, educational status, socioeconomic status, type of residence, transportation and culture”. - To include in-text citation.

Page 9: This might be due to cultural influences in the study area in which females have been still not allowed to go to health facilities for reproductive health services. It might also be due to the fact that the proportion of male and female participants in our study was different from Badewacho woreda’s (M: F, 50.9:49.1) - Provide evidence for these two points.

Page 9-10: This might be due to the fact that many of reproductive health service components (contraception) might be used when youths have perceived reproductive health risks related to sexual intercourse. - Is this your assumption? Provide evidence.

Page 10: This might probably be due to the reason that students spent their time at school during the regular health institutions’ working hours and the institution may not be possibly functional in the weekend at which the students are relatively free. Besides, since the society declares the students as they are too young to go to the health institution due to some cultural influences and visiting health institutions for particular SRH services might be thought as shameful. - Provide evidence

Page 10: Besides, the students may also perceive that long queue may let them for unnecessary exposure to the peoples who might be around the health institution and this exposure might leave frustration related to their privacy and confidentiality. Hence, the situation might be exacerbated if they think that the location of the health institution is inconvenient for them. - Provide evidence.

Page 11: Limitations - authors should discuss both the strengths and limitations in the discussion section.

Page 11: Difficult to determine the direction of causality. Factors from the service providers’ perspective, structural barriers as well as the supply was missed. - These limitations should have been discussed using proper sentences and elaborated further. Authors should also include actions to overcome those challenges.

Page 11: Factors from the service providers’ perspective, structural barriers as well as the supply was missed. - Why was it missed? This should not have happened if a comprehensive literature review was done.

Table 1: Why there were more adolescents from age group 15-19 years old?

Figure: Should be labelled as Figure 1.

Reviewer #4: General comment: The manuscript needs some minor English language corrections should it go for publication. Moreover, the research has not shown any novelty from its inception to methods of undertaking.

Abstract and background

Background of the abstract and background section of the manuscript: Authors need to justify further the rationale to the study based on existing evidences. What they put as a gap for the study is not the real gap which exists in the Ethiopian context. A number of evidences have been documented with regard to reproductive health service utilization among youths either directly or indirectly. Had it been a study referring to the challenges or quality, it would have been quite informative and having had any policy implications. However, the paper can be improved if authors are able to point out any aspects of their work (methodological or variables) which can be considered as a novel contribution to the existing evidences.

Methods: The methods section is not written in detailed manner. Since plose one considers manuscripts with rigorously worked and detailed methodology, authors need to revise the methods section so that it can explicitly depict how the samples were estimated, what procedures were followed to select the samples, how the statistical analysis was made-whether appropriate considerations were made in checking for the assumptions of various analysis made to bring about the findings. Moreover, the tool used for data collection should be discussed in terms of its sources, validity and reliability.

Authors need to discuss on the variables they have considered in the analysis. The list of variables, their categorization and which have been dropped during the bivariable analysis and why should be discussed here in the methods section.

Results: Not well structured. The very critical issue is that authors stated that they have considered health system and health care provider related factors ad their “independent factors” and failed to address these factors in the results section. Therefore, something has fallen apart here.

It would be better if authors support their “operational definitions” with either a published article or book or any other original standard material.

Discussion: The discussion is to shallow in terms of the implications of the findings for policy or programme improvements. What is the novel contribution of this research and how it is interpreted matters a lot while authors discuss their findings? In short authors need to consider a further substantiation of their “discussion “so that it can imply some thing beyond comparing and justifying for differences among findings from previous studies. For example, a number of studies reported a low utilization SRH services in the Amhara region of Ethiopia. What makes this study unique and additive to the scientific literature?

Conclusion: I would suggest authors to incorporate the recommendations written with the conclusion and the limitations to the ‘discussion” section.

Last-Authors did not follow the plose manuscript formatting style while drafting the manuscript.

6. PLOS authors have the option to publish the peer review history of their article (what does this mean?). If published, this will include your full peer review and any attached files.

Reviewer #1: No

Reviewer #2: No

Reviewer #3: No

Reviewer #4: No

---

## [Author Response · Author response to Decision Letter 0]

9 Apr 2020

Dear Editors and reviewer;

We sincerely appreciate the valuable comments and suggestions from the reviewers and editors. The thorough review helped immensely in the shaping of the manuscript. The suggestions and comments have been closely followed and revisions have been made accordingly. The following are the questions extracted from the reviewers’ comments along with our summarized responses. Thank you very much for your constrictive comments. I tried to inculcate your comments and questions as described below. The changes will be attached with.

Reviewer #1: Summary of the article

the manuscript "Youth friendly health service utilization among high and preparatory school students in Debre Tabor Town, Northwest Ethiopia" tries to assess utilization and factors affecting utilization among youths. The study may have minimal contribution to the context of the study area if the following major and minor comments are addressed by the:

1. General comment

• First of all, the issue is overlooked in various parts of the country. Some of these includes Jimma (South West Ethiopia), Harar (East ethiopia) , Anchar district (East ethiopia ) , Ethiopia (Meta-analysis ), Badwacho (Southern Ethiopia), Wolaita ( Southern Ethiopia), Hossana ( Southern Ethiopia), Asela ( South East Ethiopia), Bahir Dar ( North Ethioia ), Tigray ( North Ethiopia), Debre Birhan ( North Ethiopia), Nekemte ( West Ethiopia), Awabel district ( North west Ethiopia), Metekel Zone ( North West Ethiopia), Goba town ( South East Ethiopia), Dejene district ( North Ethiopia), Debre Markos (North Ethiopia), Woreta Town ( North West Ethiopia), Kachabira District ( Southern Ethiopia), Meachkal Distric ( North Wets Ethiopia), and Mecha district ( North west Ethiopia). If we commutate all findings, the utilization of “youth friendly sexual and reproductive health service” is very low. So what? Does this study come with a new perspective? Yes, we know the contexts particularly for these subgroup populations are different in different areas due to individual level factors, structural factors, Sociodemographic factors, and so on. Majority of the studies including the current study are done in towns and all are from youth/adolescent perspectives of the services.

• The justification “the status of these segments of population in using sexual and reproductive health services is not clearly known in the country” in the abstract section is not palatable and therefore, need another peculiar justification.

• There are issues related to the journal’s guideline like reference writing consistency

Authors Response: Accepted and corrected

2. Abstract

• Authors Response: Accepted and corrected

3. Methods section

Authors Response: The sample size was determined by using Epi info 7 statistical software by single population proportion based on following assumption 

1. Estimated proportion (p) taken from previous study done at hadiya zone on the utilization of reproductive health services among youth (29.4%)

2. Margin of error d= 5%

3. Confidence level of 95% is assumed(Z2α/2=1.96)

n = Z2α/2(P (1-P)/d2 +10% 

 Z= is the standard normal value corresponding to the desired level of confidence

d=margin of errors

p= is the estimated proportion of an attribute that is present in the population

n= Z2α/2(P (1-P)/d2 

= (1.96)2(0.29) (0.71)

 (0.05)2

 n=316, multiplying by nonresponse rate (10%) and design effect of 2 

Final sample size (nf) = [316+ (316*10%)] * 2 = 696 

Youth friendly health service utilization(Yes, No) was measured as defined on operational definition ; youth practice of utilizing reproductive health services(family planning, sexually transmitted infections, diagnosis and treatment for reproductive health issues, voluntary counselling and testing, antenatal care, abortion care, post abortion care, condom use and general health information/counselling in the past one year

It does not include diagnosis and treatment services other than reproductive health issue. Family planning was frequently facing youths.

Sampling technique 

Multi stage cluster sampling method was used to select the study participants. There were 4 high schools and 2 preparatory schools in Debre Tabor town. The number of respondents in the two schools (secondary and preparatory) was determined using probability proportionate to size allocation method. Then, 24 sections (6 sections for each high school) from a total of 81 high school sections were randomly selected. Similarly, 19 sections (12 sections from pre-1 and 7 sections from pre-2) were randomly selected from total of 64 preparatory sections. Again the students from these eligible sections were randomly selected using the list of student’s ID number as a sampling frame. Finally, the data was collected from 400 high school students and 290 preparatory students which result in the total of 690 respondents.

Simplsioo

12 sec

Pre-1

7 sec

Pre-2

6 sec

HS-1 6 sec

HS-2

 6 sec

HS-3

 6 sec

HS-4

Keys

HS- High school

Pre- Preparatory

Sec-Section

Figure 2: Schematic presentation of sampling procedures

Sexual activity: Refers to the involvement in a penetrative heterosexual intercourse within the last 12 months.

Author response: We use penetrative –to decrease recall bias ;heterosexual –we perceive homosexual is not such practiced in the study area.

Because penetrative

Awareness about SRH Services: Those knowing the availability of one or more SRH service everywhere(health institutions ,shop, drug stores ,school, market…) were reported as were reported as having awareness about the services.

How consistency of the entered data was performed? What type of consistency measures you used? Was

There the level at the acceptable range? What was it?

Consistency of data was done (1st sorting the data then compare each variables) .

4. Results Section

• Author response: the comments were accepted and corrected.

5. Reference and other issues

Author response: the comments were accepted and corrected based on Plos guideline, but there is some limitation on reference section due to insufficient internet connection in our country due to COVID-19 pandemic.

This assumption mostly works for comparative designs and has critical predictor’s nature. Your design is cross-sectional. In most cases, the cut of point of P-value is 5% ? Therefore, why is it important for you to use the cut of points? And there are comparisons in your finding with other study who were not using this assumptions. How did you see this one? Is it a just? In a bivariate logistic regression was considered to get into your multivariate analysis while your cut of point in the case of multivariate analysis is P-value<0.05 . Why? What did you do for those who said that they did not use the service?

Author response: Multivariable and multivariate are different, since there is one dependent variable which is dichotomy, we used binary logistic regression model .The appropriate model for this study is binary logistic regression model. So we can say multivariable analysis.

Scholars suggests that variable having P-Values <0.25 in bivariable logistics regression entered into multivariable analysis. And in multivariable analysis we used the cutoff point of P-value<0.05

We disseminate the finding to schools and Woreda health office, rather it is difficult identifying individuals specifically who utilized /not utilized. Based on the finding and recommendation the responsible organization will goes to action.

Why you used for your proportion ? Robust assumption? What would have been presented only the proportion?

We can put proportion without CI, but we put to make it easy for reader and CI is the base for discussion with other literatures.

Reviewer#2: Comments

Background

1. Generally you have mentioned the important point regarding issue in this study, however literature seems very brief. I would suggest to add on more

2. Paragraph 3… ‘They lack RH information, knowledge and access………service for RH’ is not clear. Maybe you could restructure the sentences

3. The acronym used in highlighting the issues not consistent as you used for SRH, RH and ARH. My suggestion is used consistency term and acronym that best applied to this study. Using full terminology rather than acronym is much better.

4. In Paragraph 6, acronym for YFS means what?

Authors response: corrected based on the comments

Methodology

1. How many population in this town and may also include how big is the town.

Author’s response Debretabor town has a total population of 55,157(FDRE, 2008). Regarding sample size calculation –addressed in reviewer 1

2. Please elaborate detail on sample size calculation; how you get 696 students participated in this study. How you did simple random sampling? How many school involved?

Author’s response regarding sample size calculation –addressed in reviewer 1.sample size was proportionally allocated for each schools. Six schools were included in the study. We found list of students from each school, we select the participants randomly by using computer method.

3. Study instrument is not clear and need further elaboration, eg what questionnaire you used, how many questions, how it is developed and does it is validated

Author’s response a total of 37 questions which consists of socio demographic, family characteristics, respondent’s awareness and sources of information about the services, and youth preferences for time and health care workers related factors. The tool was adapted from similar previous studies.

4. In writing the variables I suggest to use proper sentences. Also maybe it is very nice to put point or numbers in writing the operational definition or maybe just highlighted the variable /operational definition related to your result and discussion rather put it all as listed

Author’s response: accepted and modified

5. In ethical consideration, since your sample /participants are between age 15 to 24 years old, the ethical address should properly highlighted. Maybe you should mention at what age respondent can give consent and for those under children age (less than 18 years) how consent form was taken and should also mention others ethical issues as this topic or question that you asked give impact to subject.

Author’s response: Based on Ethiopian constitution, peoples at age of 18 years and above can give consent. For those children age (less than 18 years) assent was taken from the guardians/parents.

Result & Discussion

1. Is well presented however should report detail on response rate in this study? Maybe report the reason why they rejected to participate?

Author’s response May be :-

1. self-administered questionnaire

2. Age of the participant

3. Ignorance

2. In the discussion, paragraph 2 …(M:F, 50.9:49.1)….what does it means? Is it references?

However, maybe this study only represented youth who are in the education system since your population is not cover among those who did not enter school. This should be discuss in the limitation and maybe suggestion to improve in research.

Author’s response Male to female ratio

Conclusion & Limitation –

Author’s response: those segment of population who did not enter school were missed

 Reviewer #3:

Overall, this is a clear, concise and well-written manuscript. The topic is timely and relevant. Below are my comments:

Background

Page 1, line 1: Grammar error - "Reproductive health is 'a'..."

Author’s response; Corrected

Page 1, line 1-2: Definition of ‘reproductive health’ – this is a direct quote. Please include quotation marks.

Author’s response: accepted and corrected

Page 1, line 4: "Every year an estimated 1.7 million youths lose their lives prematurely". Were you referring SRH problems? Please specify.

Author’s response Yes ,related to reproductive health problems

Page 1, para 2: "Individuals in the age group of 15-24 years are characterized by significant physiological, psychological and social changes and making up about 20% of the world’s population, 85% lived in developing countries" - This sentence is too long and confusing. Suggest to rephrase the sentence.

Author’s response: accepted and corrected

Page 1, para 3: adolescent reproductive health (RH) - please be consistent with the abbreviation used. ARH or RH?

Author’s response: accepted and corrected

Methods

Page 2, Study period, Setting, and Population, para 1 - The study duration is not the same as stated in the abstract.

Author’s response .Accepted and corrected, due to editorial error

Why Debre Tabor town was chosen as the study setting? Please justify.

Author’s response B/c 1. No previous study conducted in this setting,

 2. There are reports by the schools and Woreda health office related to sexual and reproductive problems eg. Unintended pregnancy

 3. Utilization of reproductive health is one gap identified by students during school and home visit

Page 3, para 1 – “There are five privately owned kindergartens, 12 government and private primary schools, four government senior secondary schools (9-10), two preparatory schools, one Teachers vocational educational training, two public and one private college, and one University”- Are all these schools and colleges located in Debre Tabor town? Please be clear.

Author’s response. Yes, all are found

Page 3, para 1: "In the town, there were 4152 (2242 female and 1910 male) high schools and 2955 (1459 male and 1496 female) preparatory students" - Are high schools the same as secondary schools? This needs to be clarified from the start.

Author’s response: In our context, they are similar but for consistency we use High school as a school, which entails grades 9 and 10

Page 3, para 2 - "Night and extension students were excluded" - why were they excluded?

Author’s response ;This is b/c night and extension students:-

1. more vulnerable to this sexual and reproductive issues

2. more aware of reproductive health issue

Therefore, utilization of the service is affected

Page 3, para 2 - "A total of 696 students were participated..." - Grammar error

Author’s response :Corrected as,”A total of 696 students participated”

Page 3, para 2 - "Samples were drown..." – drawn

Author’s response; Accepted

How did the author do simple random sampling? Was random table or computer used to generate the samples?

Author’s response; Using lottery method

Why are there two subtitles? - Data Collection Methods & Data Collection Procedure. Suggest 'Data Collection’. Corrected

Author’s response; accepted and corrected

Page 3, last paragraph - "Data were collected by using structured close ended self-administered questionnaire. Data were collected by six nurses during working hours’ and supervised by three public..." - 'data were collected' used in two consecutive sentences.

Author’s response .Corrected with some rearrangement

Page 3, last paragraph: "Data were collected by six nurses during working hours..." - is it necessary to state the 'working hours'?

Author’s response .working hour is omitted

Page 3, last paragraph: The questioner was prepared in English and translated in “Amharic” for data collection and retranslated into “English” - spelling error questionnaire and you may want to use the term 'forward and backward translation' when describing the translation process.

Author’s response .Corrected as, the questionnaire was prepared in English and translated to “Amharic “.’Forward and backward translation' was done

Page 4: Dependent variable - Youth friendly health service utilization - there is no description of YFHSU. How was it measured?

Author’s response .Youth friendly health service utilization(Yes, No) was measured as defined on operational definition ; youth practice of utilizing reproductive health services(family planning, sexually transmitted infections, diagnosis and treatment for reproductive health issues, voluntary counselling and testing, antenatal care, abortion care, post abortion care, condom use and general health information/counselling in the past one year

Family characteristics - how was it measured?

Author’s response Family characteristics-it includes educational status of the mother, father (unable to read and write, able to read and write, complete primary and secondary education, diploma and above). Occupational status of the mother (house wife, merchant, government employ, others), father (farmer, merchant, government employ, others), parents alive status (alive, died)

Is the term definition for 'Reproductive health service utilization' the same as 'Youth friendly health service utilization'? Please be consistent.

Author’s response .They are almost similar despite reproductive health service utilization is broader

Page 4: Were the variables defined under 'Operational Definitions' adopted from a validated questionnaire or did the author developed their own questionnaire? This needs to be clearly stated in the manuscript.

 Author’s response The tool was adopted from previous research done, which are validated

Page 4: Youth-communication/discussion: “Youth report having talked to anyone else about one or more SRH services were categorized as having a communication/discussion about the services” - please specify the people whom they had conversations with e.g. family, teachers etc.

Author’s response .We accepted the comment, we did not consider. The operational definition deleted.

Page 5, Youth preferences: 'Preference of youth in relation to health care workers' and 'time focusing SRH services' - were these two sub-variables separated in the questionnaire?

Author’s response Yes, they were asked in separate way

How was sample size calculated? This needs to be described.

Author’s response Addressed above

Ethical considerations - was parental consent obtained? In some countries e.g. Malaysia, those less than 18 years old are considered minors and parental consent is required.

Author’s response The same is true in Ethiopia .We took parental consent for less than 18 years old 

Ethical considerations - The purpose of the study was explained to the study participants, informed written consent and assent were secured and confidentiality of the information was censured - why confidentiality was ‘censured’? Did you mean to say assured?

Author’s response .It is to mean assured

Results

Page 6, 'sources of information' - please specify. Was it sources of info on SRH?

Author’s response Yes, source of information on sexual and reproductive services

Page 7, RHSU: A total of 690 school youths were participated. – Grammar error and avoid repetition.

Author’s response .Accepted and the sentence omitted

Discussion

Page 8: “The possible explanations for the difference of the study in Harar could be justified as a higher proportion of married respondents (16.3%) in Harar may result in a higher proportion of service utilization”. - To include in-text citation.

 Author’s response .Accepted

Page 8-9: “This might be due to differences in the availability and accessibility of youth friendly health facilities, youth centers, educational status, socioeconomic status, type of residence, transportation and culture”. - To include in-text citation.

 Author’s response. Accepted

Page 9: This might be due to cultural influences in the study area in which females have been still not allowed to go to health facilities for reproductive health services. It might also be due to the fact that the proportion of male and female participants in our study was different from Badewacho woreda’s (M: F, 50.9:49.1) - Provide evidence for these two points.

Author’s response .Male to female ratio in this study was 45.5:54.5

Page 9-10: This might be due to the fact that many of reproductive health service components (contraception) might be used when youths have perceived reproductive health risks related to sexual intercourse. - Is this your assumption? Provide evidence.

Author’s response .It is known that sexual and reproductive health utilization is to prevent problems related to SRH issue.

Page 10: This might probably be due to the reason that students spent their time at school during the regular health institutions’ working hours and the institution may not be possibly functional in the weekend at which the students are relatively free. Besides, since the society declares the students as they are too young to go to the health institution due to some cultural influences and visiting health institutions for particular SRH services might be thought as shameful. - Provide evidence

Author’s response:It is known that in Ethiopia grade 1-12th will not be given from weekend period.there are cultural influence on which youth utilization of sexual and reproductive issues,so they youth afraid going to utilize the service.

Page 10: Besides, the students may also perceive that long queue may let them for unnecessary exposure to the peoples who might be around the health institution and this exposure might leave frustration related to their privacy and confidentiality. Hence, the situation might be exacerbated if they think that the location of the health institution is inconvenient for them. - Provide evidence.

Author’s response. The evidence is cultural influence on utilization of reproductive service

Page 11: Limitations - authors should discuss both the strengths and limitations in the discussion section.

Author’s response; accepted

Page 11: Difficult to determine the direction of causality. Factors from the service providers’ perspective, structural barriers as well as the supply was missed. - These limitations should have been discussed using proper sentences and elaborated further. Authors should also include actions to overcome those challenges.

Author’s response: accepted and those variables were removed

Page 11: Factors from the service providers’ perspective, structural barriers as well as the supply was missed. - Why was it missed? This should not have happened if a comprehensive literature review was done.

Author’s response This is b/c our study was limited only student’s perecipective

Table 1: Why there were more adolescents from age group 15-19 years old?

Author’s response This is b/c the participants were from grade 9-12, in which most of them were in the age group of 15-19

Figure: Should be labelled as Figure 1.

Author’s response Corrected in the main document

Reviewer #4:

 General comment: The manuscript needs some minor English language corrections should it go for publication. Moreover, the research has not shown any novelty from its inception to methods of undertaking.

Abstract and background

Background of the abstract and background section of the manuscript: Authors need to justify further the rationale to the study based on existing evidences. What they put as a gap for the study is not the real gap which exists in the Ethiopian context. A number of evidences have been documented with regard to reproductive health service utilization among youths either directly or indirectly. Had it been a study referring to the challenges or quality, it would have been quite informative and having had any policy implications. However, the paper can be improved if authors are able to point out any aspects of their work (methodological or variables) which can be considered as a novel contribution to the existing evidences.

Author’s response.accepted and included in the revised manuscript

Methods: The methods section is not written in detailed manner. Since plose one considers manuscripts with rigorously worked and detailed methodology, authors need to revise the methods section so that it can explicitly depict how the samples were estimated, what procedures were followed to select the samples, how the statistical analysis was made-whether appropriate considerations were made in checking for the assumptions of various analysis made to bring about the findings. Moreover, the tool used for data collection should be discussed in terms of its sources, validity and reliability.

Authors need to discuss on the variables they have considered in the analysis. The list of variables, their categorization and which have been dropped during the bivariable analysis and why should be discussed here in the methods section.

Author’s response: We think it has been shown in the table 4,to avoid redundancy we leave it

Results: Not well structured. The very critical issue is that authors stated that they have considered health system and health care provider related factors ad their “independent factors” and failed to address these factors in the results section. Therefore, something has fallen apart here.

It would be better if authors support their “operational definitions” with either a published article or book or any other original standard material.

Author’s response .Accepted and modified. Primarily we tried to consider but we missed to collect data on health system and health care provider related factors.

Discussion: The discussion is to shallow in terms of the implications of the findings for policy or programme improvements. What is the novel contribution of this research and how it is interpreted matters a lot while authors discuss their findings? In short authors need to consider a further substantiation of their “discussion “so that it can imply some thing beyond comparing and justifying for differences among findings from previous studies. For example, a number of studies reported a low utilization SRH services in the Amhara region of Ethiopia. What makes this study unique and additive to the scientific literature?

Author’s response: the contribution of the study is: - shows barriers in the study setting, baseline for implementation of youth friendly health services

Conclusion: I would suggest authors to incorporate the recommendations written with the conclusion and the limitations to the ‘discussion” section.

Author’s response. Accepted

Last-Authors did not follow the plose manuscript formatting style while drafting the manuscript.

Author’s response. Accepted and corrected per the guideline

---

## [Editor Report · Decision Letter 1]

24 Apr 2020

PONE-D-19-33641R1

Youth friendly health service utilization among high and preparatory school students in Debre Tabor Town, Northwest Ethiopia.

PLOS ONE

Dear Mr. Minuye,

Thank you for submitting your manuscript to PLOS ONE. After careful consideration, we feel that it has merit but does not fully meet PLOS ONE’s publication criteria as it currently stands. Therefore, we invite you to submit a revised version of the manuscript that addresses the points raised during the review process.

We would appreciate receiving your revised manuscript by Jun 08 2020 11:59PM. To enhance the reproducibility of your results, we recommend that if applicable you deposit your laboratory protocols in protocols.io, where a protocol can be assigned its own identifier (DOI) such that it can be cited independently in the future. For instructions see: http://journals.plos.org/plosone/s/submission-guidelines#loc-laboratory-protocols

We look forward to receiving your revised manuscript.

Kind regards,

Julie Maslowsky, PhD

Academic Editor

PLOS ONE

Additional Editor Comments (if provided):

Dear authors,

Before I make a decision on your revised manuscript, I would like to request that you please reformat and resubmit the "response to reviewers". For each reviewer comment, please include a detailed account of how and where you responded to the comment. For example, in Reviewer 1's first comment, there are a number of suggestions and questions to the author. The author's response indicates only "accepted and corrected". I would ask that the authors please include a detailed explanation of the changes they made to the manuscript in response to this comment, including the section of the manuscript and page number. Please also include the original reviewer comment as well as your response. For example, Reviewer 1's comment about the methods section has been deleted. It is difficult for reviewers and the editor to evaluate your responses to the reviewers without having the reviewers' comments included with the response. Please resubmit the manuscript with the revised "response to reviewers".

Thank you.

---

## [Author Response · Author response to Decision Letter 1]

29 Jul 2020

Dear Editors and reviewer;

We sincerely appreciate the valuable comments and suggestions from the reviewers and editors. The thorough review helped immensely in the shaping of the manuscript. The suggestions and comments have been closely followed and revisions have been made accordingly. The following are the questions extracted from the reviewers’ comments along with our summarized responses. Thank you very much for your constrictive comments. We tried to inculcate your comments and questions as described below. The changes will be attached with.

Title: Youth friendly health service utilization among high and preparatory school students in Debre Tabor Town, Northwest Ethiopia.

Authors:

Amare Simegn (amaresimegn99@gmail.com)

Telake Azale (Telakea@yahoo.com)

Abebaw Addis (Abebaw.addis@gmail.com)

Mulugeta Dile (muliedile@gmail.com)

Yitayal Ayalew (ayalewyitayal@gmail.com)

Biniam Minuye (biniamminuye@yahoo.com

Version 1-date June 1 2020

Point by point Author’s response to reviews

Reviewer#1:

Reviewer 1 comments and questions: The manuscript "Youth friendly health service utilization among high and preparatory school students in Debre Tabor Town, Northwest Ethiopia" tries to assess utilization and factors affecting utilization among youths. The study may have minimal contribution to the context of the study area if the following major and minor comments are addressed by the:

1.General comment: First of all, the issue is overlooked in various parts of the country. Some of these includes Jimma (South West Ethiopia), Harar (East ethiopia) , Anchar district (East ethiopia ) , Ethiopia (Meta-analysis ), Badwacho (Southern Ethiopia), Wolaita ( Southern Ethiopia), Hossana ( Southern Ethiopia), Asela ( South East Ethiopia), Bahir Dar ( North Ethioia ), Tigray ( North Ethiopia), Debre Birhan ( North Ethiopia), Nekemte ( West Ethiopia), Awabel district ( North west Ethiopia), Metekel Zone ( North West Ethiopia), Goba town ( South East Ethiopia), Dejene district ( North Ethiopia), Debre Markos (North Ethiopia), Woreta Town ( North West Ethiopia), Kachabira District ( Southern Ethiopia), Meachkal Distric ( North Wets Ethiopia), and Mecha district ( North west Ethiopia). If we commutate all findings, the utilization of “youth friendly sexual and reproductive health service” is very low. So what? Does this study come with a new perspective? Yes, we know the contexts particularly for these subgroup populations are different in different areas due to individual level factors, structural factors, Sociodemographic factors, and so on. Majority of the studies including the current study are done in towns and all are from youth/adolescent perspectives of the services.

• The justification “the status of these segments of population in using sexual and reproductive health services is not clearly known in the country” in the abstract section is not palatable and therefore, need another peculiar justification.

• There are issues related to the journal’s guideline like reference writing consistency

Authors Response: Thank you for your critical view of our paper

Despite this type of study is conducted in different parts of Ethiopia, Youth continue to fall victim to sexual and reproductive health problems, its consequences. Despite, reproductive health needs of youth had been supported by government and non-governmental organizations to increase access to sexual and reproductive health services by young people. All these efforts have not been felt across the Ethiopian learning institutions as is evidenced by persistent reproductive health problems and challenges of the youth including in the study setting.

Reviewer 1 comments: Use Google scholar. There are ample amount of research as I put the summary of me review. Would you like to present other justification that this study contributes to

Authors Response: as per the reviewer comment the abstract section has been rewrite as “Youth continue to fall victim to sexual and reproductive health problems, its consequences. Despite, reproductive health needs of youth had been supported by government and non-governmental organizations to increase access to sexual and reproductive health services by young people. All these efforts have not been felt across the Ethiopian learning institutions as is evidenced by persistent reproductive health problems and challenges of the youth including in the study setting.”

Paragraph 1 line 2 on background section. “Reproductive health problems faced by Ethiopian youth are tremendous .The status of these segments of population in using sexual and reproductive health services is not clearly known in the country.” Was deleted

Reviewer 1 comments and question: What 28.8% in the result subsection of the abstract?

Authors Response: 28.8% indicates overall utilization of reproductive health, based on the comment the confidence interval was deleted.On conclusion section we add the following recommendation, Future researcher should address segment of population who does not enter school.

Reviewer 1 Comments and questions: on paragraph 2 line 2 what 15-24?

Author response: based on the comment we rewrite the paragraph as “Youth are characterized by significant physiological, psychological and social changes. This shares about 20% of the world’s population. More than one third of the Ethiopian population is aged between 15-24,mostly vulnerable to a range of sexual and reproductive health problems

In the paragraph 2 line 1, in the age group of 15-24 years -deleted

In paragraph 2 line 3 & 4 and making up and population 85% lived in developing countries –deleted

Reviewer 1 Comments and questions: paragraph 5 line 6-7, However, the effects of all these efforts have not been well understood across the Ethiopian high schools and preparatory schools as is evidenced by persistent reproductive health problems and challenges to the youth?? Still better than stated in your Abstract section, But what make different from for example “Debre Markos” in the same region ??

Author response: Based on our understanding, it needs coherent with the following section.so we tried to rewrite .based on the comments we rewrite the abstract section

Paragraph 6 line first sentence: There is limited evidence of utilizing SRH services and the existing data didn’t fully assess the level and factors for RH service utilization in Debre Tabor town and the resulting information can bridge the gap of the existed little knowledge in the setting -deleted.

And we add the following in place “Utilization of reproductive health services is the key for the improvement in the quality of life of the youth. Behaviors formed and choices made during adolescence period. Despite different studies conducted in Ethiopia, existing data didn’t fully assess the factors which affect service utilization.”

3. Method section

Reviewer 1 Question: How this sample statistically determined? What were your assumptions?

Authors Response: Method section-replaced by materials and methods

The sample size was determined by using Epi info 7 statistical software by single population proportion based on following assumption 

1. Estimated proportion (p) taken from previous study done at hadiya zone on the utilization of reproductive health services among youth (29.4%)

2. Margin of error d= 5%

3. Confidence level of 95% is assumed(Z2α/2=1.96)

n = Z2α/2(P (1-P)/d2 +10% 

 Z= is the standard normal value corresponding to the desired level of confidence

d=margin of errors

p= is the estimated proportion of an attribute that is present in the population

n= Z2α/2(P (1-P)/d2 

= (1.96)2(0.29) (0.71)

 (0.05)2

 n=316, multiplying by nonresponse rate (10%) and design effect of 2 

Final sample size (nf) = [316+ (316*10%)] * 2 = 696 

Sampling technique -is added in revised manuscript 

Multi stage cluster sampling method was used to select the study participants. There were 4 high schools and 2 preparatory schools in Debre Tabor town. The number of respondents in the two schools (secondary and preparatory) was determined using probability proportionate to size allocation method. Then, 24 sections (6 sections for each high school) from a total of 81 high school sections were randomly selected. Similarly, 19 sections (12 sections from pre-1 and 7 sections from pre-2) were randomly selected from total of 64 preparatory sections. Again the students from these eligible sections were randomly selected using the list of student’s ID number as a sampling frame. Finally, the data was collected from 400 high school students and 290 preparatory students which result in the total of 690 respondents.

Simplsioo

12 sec

Pre-1

7 sec

Pre-2

6 sec

HS-1 6 sec

HS-2

 6 sec

HS-3

 6 sec

HS-4

Keys

HS- High school

Pre- Preparatory

Sec-Section

Figure 1: Schematic presentation of sampling procedures

Reviewer comment and Question: How Youth friendly health service utilization was measured?

Authors Response :Youth friendly health service utilization(Yes, No) was measured as defined on operational definition ; youth practice of utilizing reproductive health services(family planning, sexually transmitted infections, diagnosis and treatment for reproductive health issues, voluntary counselling and testing, antenatal care, abortion care, post abortion care, condom use and general health information/counselling in the past one year.

Reviewer 1 comment and Question: Does it include other “diagnosis and treatment” services ? Can a medical diagnosis and treatment other than sexual health issues be an indicator for SRH utilization? If so, how youths at least seek medical services at least once in a year? 28.8% is very low? That is why such study may not come up with a different perspective? What sexual issues are frequently facing youths?

Authors Response: It does not include diagnosis and treatment services other than reproductive health issue. Family planning was frequently facing youths.

Reviewer 1 comment: Provide a heading and write using standard writing?

Authors Response: Based on the comment we made independent variable as a heading

 In operational section paragraph 1 line 8 : Utilization –replaced by is utilized

Reviewer 1 comment and Question on: Sexual activity: Refers to the involvement in a penetrative heterosexual intercourse?? within the last 12 months.

Author response: We use penetrative –to decrease recall bias; heterosexual –we perceive homosexual is not such practiced in the study area.

Reviewer 1 comment and Question on: Awareness about SRH Services: Those knowing the availability of one or more SRH service in where?? Were reported as having awareness about the services.

Author response: Awareness about SRH Services: Those knowing the availability of one or more SRH service everywhere(health institutions ,shop, drug stores ,school, market…) were reported as were reported as having awareness about the services.

Reviewer 1 comment and Question: How consistency of the entered data was performed? What type of consistency measures you used? Was there the level at the acceptable range? What was it?

Author response: Consistency of data was done (1st sorting the data then compare each variables) .To maintain the consistency after data double entry, the entered data were cross checked by researcher and evaluator and if there inconsistency the researcher goes back to original data and correct accordingly after search for the possible errors. Furthermore, Cronbach alpha were used to measure the consistency of the data with the value of 7.9. The acceptable range of cronbach alpha is 0.6-0.9

Reviewer 1 comment and Question: This assumption mostly works for comparative designs and has critical predictor’s nature. Your design is cross-sectional. In most cases, the cut of point of P-value is 5% ? Therefore, why is it important for you to use the cut of points? And there are comparisons in your finding with other study who were not using this assumptions. How did you see this one? Is it a just?

Author response: It is to say multivariable analysis in binary logistic regression model. To show significant associations of outcome and independent variables the value of p is either less or equal to 0.05. A small p-value (typically ≤ 0.05) indicates strong evidence against the null hypothesis, so you reject the null hypothesis.

Reviewer 1 comment and Question: In a bivariate logistic regression was considered to get into your multivariate analysis while your cut of point in the case of multivariate analysis is P-value<0.05. Why? What did you do for those who said that they did not use the service?

Author response: Multivariable and multivariate are different, since there is one dependent variable which is dichotomy, we used binary logistic regression model .The appropriate model for this study is binary logistic regression model. So we can say multivariable analysis. Scholars suggests that variable having P-Values <0.25 in bivariable logistics regression entered into multivariable analysis. And in multivariable analysis we used the cutoff point of P-value<0.05.

Reviewer 1 comment and Question: What did you do for those who said that they did not use the service?

Author response: We disseminate the finding to schools and Woreda health office, rather it is difficult identifying individuals specifically who utilized /not utilized. Based on the finding and recommendation the responsible organization will goes to action.

In the ethical section assent was replaced by children less than 18 years old and Censured-replaced by ensured

Result section

Reviewer 1 comment and Question: Why you used for your proportion? Robust assumption? What would have been presented only the proportion? Multivariate?

Author response: We can put proportion without CI, but we put to make it easy for reader and CI is the base for discussion with other literatures. Based on the comment confidence interval was omitted. Multivariate replaced by multivariable –it was editorial error

Discussion section

Reviewer 1 comment and Question: In discussion section paragraph 2 line 5(M: F, 50.9:49.1)??

Author response: the short form (M: F, 50.9:49.1) is to mean male to female ratio

Reviewer comment and Question: in conclusion?

Author response: the world “in Conclusion “in the first section was deleted as per the comment

Reference section:

Reviewer 1 comment and Question: Check Authors guideline and follow consistent referencing style?

Author response: references were corrected based on Plos guideline

Reviewer#2: Comments

Reviewer 2 comments and questions: generally you have mentioned the important point regarding issue in this study, however literature seems very brief. I would suggest to add on more

Author’s response: thanks for your valuable comments and questions related to the paper. We tried to respond point by point for your comments and questions. Based on the comment given we rewrite literatures to make it more meangiful and we add the following “Utilization of reproductive health services is the key for the improvement in the quality of life of the youth. Behaviors formed and choices made during adolescence period. Despite different studies conducted in Ethiopia, existing data didn’t fully assess the factors which affect service utilization.

Reviewer 2 comments and questions : Paragraph 3… ‘They lack RH information, knowledge and access………service for RH’ is not clear. Maybe you could restructure the sentences

Author’s response: we accepted the comment and restructured the whole paragraph 3 as “Focusing on adolescent reproductive health is both a challenge and an opportunity for health care providers. Adolescence generally is a healthy period of life, many youths are less informed on reproductive health services, less experienced, and less comfortable accessing health services for reproductive issues than adults”

Reviewer 2 comments and questions :The acronym used in highlighting the issues not consistent as you used for SRH, RH and ARH. My suggestion is used consistency term and acronym that best applied to this study. Using full terminology rather than acronym is much better.

Author’s response: As per the comment we used the acronym RH consistently

Reviewer 2 comments and questions: In Paragraph 6, acronym for YFS means what?

Author’s response: it is to mean Youth friendly service

Methodology section

Reviewer 2 comments and questions. How many population in this town and may also include how big is the town.

Author’s response ; Debretabor town has a total population of 55,157(FDRE, 2008).

Reviewer 2 comments and questions: Please elaborate detail on sample size calculation; how you get 696 students participated in this study. How you did simple random sampling? How many school involved?

Author’s response: regarding sample size calculation –addressed in reviewer 1.sample size was proportionally allocated for each schools. Six schools were included in the study. We found list of students from each school, we select the participants randomly by using computer method.

Reviewer 2 comments and questions: Study instrument is not clear and need further elaboration, eg what questionnaire you used, how many questions, how it is developed and does it is validated

Author’s response a total of 37 questions which consists of socio demographic, family characteristics, respondent’s awareness and sources of information about the services, and youth preferences for time and health care workers related factors. The tool was adapted from similar previous studies.

Reviewer 2 comments and questions: In writing the variables I suggest to use proper sentences. Also maybe it is very nice to put point or numbers in writing the operational definition or maybe just highlighted the variable /operational definition related to your result and discussion rather put it all as listed

Author’s response: we highlight the operational variables and Youth communication/discussion is deleted

Reviewer 2 comments and questions: In ethical consideration, since your sample /participants are between age 15 to 24 years old, the ethical address should properly highlighted. Maybe you should mention at what age respondent can give consent and for those under children age (less than 18 years) how consent form was taken and should also mention others ethical issues as this topic or question that you asked give impact to subject.

Author’s response: Based on Ethiopian constitution, peoples at age of 18 years and above can give consent. For those children age (less than 18 years) assent (written consent) was taken from the guardians/parents.

 Result & Discussion

Reviewer 2 comments and questions: Is well presented however should report detail on response rate in this study? Maybe report the reason why they rejected to participate?

Author’s response May be :-

1. self-administered questionnaire

2. Age of the participant

3. Ignorance

Reviewer comments and questions: In the discussion, paragraph 2 … (M:F, 50.9:49.1)….what does it means? Is it references?

However, maybe this study only represented youth who are in the education system since your population is not cover among those who did not enter school. This should be discuss in the limitation and maybe suggestion to improve in research.

Author’s response :Male to female ratio

Conclusion & Limitation –

Author’s response: those segment of population who did not enter school were missed

Reviewer #3:

Overall, this is a clear, concise and well-written manuscript. The topic is timely and relevant. Below are my comments:

Background

Reviewer 3 comments and questions: Page 1, line 1: Grammar error - "Reproductive health is 'a'..."

Author’s response; thank you very much for your view of the paper. Corrected as per the comment as “Reproductive health is,” a state of…..”

Reviewer 3 comments and questions: Page 1, line 1-2: Definition of ‘reproductive health’ – this is a direct quote. Please include quotation marks.

Author’s response: Based on the comment we put the following under quotation “state of complete physical, mental and social well-being and not merely the absence of disease/infirmity, in all matters relating to the reproductive system.

Reviewer 3 comments and questions: Page 1, line 4: "Every year an estimated 1.7 million youths lose their lives prematurely". Were you referring SRH problems? Please specify.

Author’s response Yes ,related to reproductive health problems

Reviewer 3 comments and questions :Page 1, para 2: "Individuals in the age group of 15-24 years are characterized by significant physiological, psychological and social changes and making up about 20% of the world’s population, 85% lived in developing countries" - This sentence is too long and confusing. Suggest to rephrase the sentence.

Author’s response: Paragraph 2 page 1-Rephrased as “Youth are characterized by significant physiological, psychological and social changes. This shares about 20% of the world’s population. More than one third of the Ethiopian population is aged between 15-24, mostly vulnerable to a range of sexual and reproductive health problems.

Reviewer 3 comments and questions: Page 1, para 3: adolescent reproductive health (RH) - please be consistent with the abbreviation used. ARH or RH?

Author’s response: Corrected and RH acronyms used consistently throughout the document

Reviewer 3 comments and questions: Page 2, Study period, Setting, and Population, para 1 - The study duration is not the same as stated in the abstract.

Author’s response .this is due to editorial error

Reviewer 3 comments and questions : Why Debre Tabor town was chosen as the study setting? Please justify.

Author’s response B/c 1. No previous study conducted in this setting,

 2. There are reports by the schools and Woreda health office related to sexual and reproductive problems eg. Unintended pregnancy

 3. Utilization of reproductive health is one gap identified by students during school and home visit

Reviewer 3 comments and questions :Page 3, para 1 – “There are five privately owned kindergartens, 12 government and private primary schools, four government senior secondary schools (9-10), two preparatory schools, one Teachers vocational educational training, two public and one private college, and one University”- Are all these schools and colleges located in Debre Tabor town? Please be clear.

Author’s response. Yes, all are found

Reviewer 3 comments and questions: Page 3, para 1: "In the town, there were 4152 (2242 female and 1910 male) high schools and 2955 (1459 male and 1496 female) preparatory students" - Are high schools the same as secondary schools? This needs to be clarified from the start.

Author’s response: In our context, they are similar but for consistency we use High school as a school, which entails grades 9 and 10

Reviewer comments and questions: Page 3, para 2 - "Night and extension students were excluded" - why were they excluded?

Author’s response; this is b/c night and extension students:-

1. more vulnerable to this sexual and reproductive issues

2. more aware of reproductive health issue

Therefore, utilization of the service is affected

Reviewer 3 comments and questions : Page 3, para 2 - "A total of 696 students were participated..." - Grammar error

Author’s response: Corrected as, “A total of 696 students participated”

Reviewer 3 comments and questions: Page 3, para 2 - "Samples were drown..."

Author’s response; this is grammatical error and corrected as – drawn

Reviewer comments and questions: How did the author do simple random sampling? Was random table or computer used to generate the samples?

Author’s response; neither of them, we use lottery method

Reviewer 3 comments and questions: Why are there two subtitles? - Data Collection Methods & Data Collection Procedure. Suggest 'Data Collection’.

Author’s response; per the comment we correct as data collection in place of data collection methods and procedure

Reviewer 3 comments and questions: Page 3, last paragraph - "Data were collected by using structured close ended self-administered questionnaire. Data were collected by six nurses during working hours’ and supervised by three public..." - 'data were collected' used in two consecutive sentences.

Author’s response .Corrected as “Data were collected by using structured close ended self-administered questionnaire. A total of 37 questions which consists of socio demographic, family characteristics, respondent’s awareness and sources of information about the services, health system factors and youth preferences for time and health care workers related factors.

 Reviewer 3 comments and questions: Page 3, last paragraph: "Data were collected by six nurses during working hours..." - is it necessary to state the 'working hours'?

Author’s response .working hour is omitted and written as Data were collected by six nurses and supervised by three public health professionals. 

Reviewer 3 comments and questions : Page 3, last paragraph: The questioner was prepared in English and translated in “Amharic” for data collection and retranslated into “English” - spelling error questionnaire and you may want to use the term 'forward and backward translation' when describing the translation process.

Author’s response .Corrected as, the questionnaire was prepared in English and translated to “Amharic “.’Forward and backward translation' was done

Reviewer 3 comments and questions: Page 4: Dependent variable - Youth friendly health service utilization - there is no description of YFHSU. How was it measured?

Author’s response .Youth friendly health service utilization(Yes, No) was measured as defined on operational definition ; youth practice of utilizing reproductive health services(family planning, sexually transmitted infections, diagnosis and treatment for reproductive health issues, voluntary counselling and testing, antenatal care, abortion care, post abortion care, condom use and general health information/counselling in the past one year

Reviewer 3 comments and questions :Family characteristics - how was it measured?

Author’s response Family characteristics-it includes educational status of the mother, father (unable to read and write, able to read and write, complete primary and secondary education, diploma and above). Occupational status of the mother (house wife, merchant, government employ, others), father (farmer, merchant, government employ, others), parents alive status (alive, died)

Reviewer 3 comments and questions: Is the term definition for 'Reproductive health service utilization' the same as 'Youth friendly health service utilization'? Please be consistent.

Author’s response .They are almost similar despite reproductive health service utilization is broader

Reviewer 3 comments and questions :Page 4: Were the variables defined under 'Operational Definitions' adopted from a validated questionnaire or did the author developed their own questionnaire? This needs to be clearly stated in the manuscript.

 Author’s response The tool was adopted from previous research done, which are validated

Reviewer 3 comments and questions : Page 4: Youth-communication/discussion: “Youth report having talked to anyone else about one or more SRH services were categorized as having a communication/discussion about the services” - please specify the people whom they had conversations with e.g. family, teachers etc.

Author’s response .We accepted the comment, we did not consider. The operational definition deleted.

Reviewer 3 comments and questions :Page 5, Youth preferences: 'Preference of youth in relation to health care workers' and 'time focusing SRH services' - were these two sub-variables separated in the questionnaire?

Author’s response Yes, they were asked in separate way

Reviewer 3 comments and questions :How was sample size calculated? This needs to be described.

Author’s response Addressed above

Reviewer 3 comments and questions :Ethical considerations - was parental consent obtained? In some countries e.g. Malaysia, those less than 18 years old are considered minors and parental consent is required.

Author’s response The same is true in Ethiopia .We took parental consent for less than 18 years old 

Reviewer 3 comments and questions :Ethical considerations - The purpose of the study was explained to the study participants, informed written consent and assent were secured and confidentiality of the information was censured - why confidentiality was ‘censured’? Did you mean to say assured?

Author’s response .It is to mean assured

Results

Reviewer comments and questions: Page 6, 'sources of information' - please specify. Was it sources of info on SRH?

Author’s response Yes, source of information on sexual and reproductive services

Page 7, RHSU: A total of 690 school youths were participated. – Grammar error and avoid repetition.Accepted and the sentence omitted

Discussion

Reviewer 3 comments and questions: Page 8: “The possible explanations for the difference of the study in Harar could be justified as a higher proportion of married respondents (16.3%) in Harar may result in a higher proportion of service utilization”. - To include in-text citation.

 Author’s response .we include the citation as per the comment

Reviewer 3 comments and questions :Page 8-9: “This might be due to differences in the availability and accessibility of youth friendly health facilities, youth centers, educational status, socioeconomic status, type of residence, transportation and culture”. - To include in-text citation.

 Author’s response. Accepted and placed in text citation

Reviewer comments and questions: Page 9: This might be due to cultural influences in the study area in which females have been still not allowed to go to health facilities for reproductive health services. It might also be due to the fact that the proportion of male and female participants in our study was different from Badewacho woreda’s (M: F, 50.9:49.1) - Provide evidence for these two points.

Author’s response .Male to female ratio in this study was 45.5:54.5it is researcher’s opinion

Reviewer comments and questions: Page 9-10: This might be due to the fact that many of reproductive health service components (contraception) might be used when youths have perceived reproductive health risks related to sexual intercourse. - Is this your assumption? Provide evidence.

Author’s response .It is known that sexual and reproductive health utilization is to prevent problems related to SRH issue.

Reviewer 3 comments and questions :Page 10: This might probably be due to the reason that students spent their time at school during the regular health institutions’ working hours and the institution may not be possibly functional in the weekend at which the students are relatively free. Besides, since the society declares the students as they are too young to go to the health institution due to some cultural influences and visiting health institutions for particular SRH services might be thought as shameful. - Provide evidence

Author’s response: It is known that in Ethiopia grade 1-12th will not be given from weekend period.there are cultural influence on which youth utilization of sexual and reproductive issues,so they youth afraid going to utilize the service.

Reviewer 3 comments and questions: Page 10: Besides, the students may also perceive that long queue may let them for unnecessary exposure to the peoples who might be around the health institution and this exposure might leave frustration related to their privacy and confidentiality. Hence, the situation might be exacerbated if they think that the location of the health institution is inconvenient for them. - Provide evidence.

Author’s response. The evidence is cultural influence on utilization of reproductive service

Reviewer 3 comments and questions: Page 11: Limitations - authors should discuss both the strengths and limitations in the discussion section.

Author’s response; we tried to incorporate the strength and limitation of the study on discussion section .we add “Therefore, educating female students to go to health facilities for SRH services, creating awareness on the preventive aspects of SRH problems and advocating SRH service discussion among themselves and with others are important. Since the study is crossectional it does not show cause effect relationship. Those segment of population who did not enter school were missed. Factors from the service providers’ perspective, structural barriers as well as the supply was missed” on the discussion section

Reviewer 3 comments and questions: Page 11: Difficult to determine the direction of causality. Factors from the service providers’ perspective, structural barriers as well as the supply was missed. - These limitations should have been discussed using proper sentences and elaborated further. Authors should also include actions to overcome those challenges.

Author’s response: accepted and those variables were removed

Reviewer 3 comments and questions: Page 11: Factors from the service providers’ perspective, structural barriers as well as the supply was missed. - Why was it missed? This should not have happened if a comprehensive literature review was done.

Author’s response this is b/c our study was limited only student’s perspective

Reviewer 3 comments and questions: Table 1: Why there were more adolescents from age group 15-19 years old?

Author’s response : this is b/c the participants were from grade 9-12, in which most of them were in the age group of 15-19

Reviewer 3 comments and questions :Figure: Should be labelled as Figure 1.

Author’s response Corrected in the main document as; Figure 1: Schematic presentation of sampling procedures

Reviewer #4:

Reviewer 4 comments and questions: General comment: The manuscript needs some minor English language corrections should it go for publication. Moreover, the research has not shown any novelty from its inception to methods of undertaking.

Author’s response: we would like to thank you for your critical view, suggestions and questions for the betterment of the manuscript. The manuscript was edited by different personal and we tried to restructure the manuscript after critical review and comments from reviewers.

Reviewer 4 comments and questions: Background of the abstract and background section of the manuscript: Authors need to justify further the rationale to the study based on existing evidences. What they put as a gap for the study is not the real gap which exists in the Ethiopian context. A number of evidences have been documented with regard to reproductive health service utilization among youths either directly or indirectly. Had it been a study referring to the challenges or quality, it would have been quite informative and having had any policy implications. However, the paper can be improved if authors are able to point out any aspects of their work (methodological or variables) which can be considered as a novel contribution to the existing evidences.

Author’s response. As per the comment given authors need to address the Accepted and included in the revised manuscript

Reviewer 4 comments and questions: Methods: The methods section is not written in detailed manner. Since plose one considers manuscripts with rigorously worked and detailed methodology, authors need to revise the methods section so that it can explicitly depict how the samples were estimated, what procedures were followed to select the samples, how the statistical analysis was made-whether appropriate considerations were made in checking for the assumptions of various analysis made to bring about the findings. Moreover, the tool used for data collection should be discussed in terms of its sources, validity and reliability.

Authors need to discuss on the variables they have considered in the analysis. The list of variables, their categorization and which have been dropped during the bivariable analysis and why should be discussed here in the methods section.

Author’s response: We think it has been shown in the table 4, some section in the methodology section like sample size calculation, sampling procedure ,data collection methods were missed in the previous manuscript but added here and explained in response for other reviewer, to avoid redundancy we leave it

Reviewer 4 comments and questions: Not well structured. The very critical issue is that authors stated that they have considered health system and health care provider related factors ad their “independent factors” and failed to address these factors in the results section. Therefore, something has fallen apart here. It would be better if authors support their “operational definitions” with either a published article or book or any other original standard material.

Author’s response .Primarily we tried to consider but we missed to collect data on health system and health care provider related factors which we face difficult to find.

Reviewer 4 comments and questions: The discussion is to shallow in terms of the implications of the findings for policy or programme improvements. What is the novel contribution of this research and how it is interpreted matters a lot while authors discuss their findings? In short authors need to consider a further substantiation of their “discussion “so that it can imply something beyond comparing and justifying for differences among findings from previous studies. For example, a number of studies reported a low utilization SRH services in the Amhara region of Ethiopia. What makes this study unique and additive to the scientific literature?

Author’s response: the contribution of the study is: - shows barriers in the study setting, baseline for implementation of youth friendly health services in the study setting. Identifying this gap helps for future implementation of services in sexual and reproductive areas

Reviewer 4 comments and questions: I would suggest authors to incorporate the recommendations written with the conclusion and the limitations to the ‘discussion” section.

Author’s response. As per the comment given we incorporate in discussion section

Reviewer 4 comments and questions -Authors did not follow the plose manuscript formatting style while drafting the manuscript.

Author’s response. We tried to amend as per the guideline

Sincerely yours,

Biniam Minuye (on behalf of the authors)

---

## [Decision Letter · Decision Letter 2]

4 Sep 2020

PONE-D-19-33641R2

Youth friendly health service utilization among high and preparatory school students in Debre Tabor Town, Northwest Ethiopia. A cross-sectional study design

PLOS ONE

Dear Dr. Minuye,

Thank you for submitting your manuscript to PLOS ONE. After careful consideration, we feel that it has merit but does not fully meet PLOS ONE’s publication criteria as it currently stands. Therefore, we invite you to submit a revised version of the manuscript that addresses the points raised during the review process.

We look forward to receiving your revised manuscript.

Kind regards,

Julie Maslowsky, PhD

Academic Editor

PLOS ONE

Additional Editor Comments (if provided):

Thank you for your revised manuscript. The reviewers have asked for some additional edits to this paper. Please respond to all reviewer comments.

Reviewers' comments:

Reviewer's Responses to Questions

**Comments to the Author**

1. If the authors have adequately addressed your comments raised in a previous round of review and you feel that this manuscript is now acceptable for publication, you may indicate that here to bypass the “Comments to the Author” section, enter your conflict of interest statement in the “Confidential to Editor” section, and submit your "Accept" recommendation.

Reviewer #1: All comments have been addressed

Reviewer #3: All comments have been addressed

2. Is the manuscript technically sound, and do the data support the conclusions?

Reviewer #1: Partly

Reviewer #3: Yes

3. Has the statistical analysis been performed appropriately and rigorously? 

Reviewer #1: Yes

Reviewer #3: Yes

4. Have the authors made all data underlying the findings in their manuscript fully available?

Reviewer #1: Yes

Reviewer #3: Yes

5. Is the manuscript presented in an intelligible fashion and written in standard English?

Reviewer #1: No

Reviewer #3: Yes

6. Review Comments to the Author

Reviewer #1: The following important points may improve the manuscript

1. revise abstract section according to comments provided in track change file

2. extensive language editing may benefit the article

3. Proper leveling of headings and subheadings of section of the manuscript may be important

4. Sampling techniques need some revision as per comments provided in track change file

Reviewer #3: Title: Youth friendly health service utilization among high and preparatory school students in Debre Tabor Town, Northwest Ethiopia. A cross-sectional study design

Reviewer: The term ' sexual and reproductive' should be added to the title to reflect the content of the manuscript. I suggest 'Youth friendly sexual and reproductive health service utilization among high and preparatory school students in Debre Tabor Town, Northwest Ethiopia. A cross sectional study.

Sample size calculation.

Reviewer: This should be written in one paragraph. The formula is not necessary.

Operational definitions.

Reviewer: The variables could be defined under the variables section.

Ethical clearance was obtained from Gondar University, College of Medical and Health Sciences, Institutional Health Research Ethics Review Committee (IHRERC).

Reviewer: Please provide the ethics approval/ registration number.

Ethics: The purpose of the study was explained to the study participants, informed written consent and assent were secured for children less than 18 years and confidentiality of the information was ensured.

Ethics section.

Reviewer: Was parental consent taken for those below 18 years old? If yes, this needs to be written in the manuscript.

294 (42.6%) of students responded that the health care facility found in their respective residences was not suitable to use the services due to the mistreating health care providers, long-distance, and unfavorable service hours.

Reviewer: Avoid begin a sentence with a number.

However, 90 (13.0%) of the respondents reported that the health care providers were mistreated.

Reviewer: Shouldn't this be the other way round? The youths were mistreated instead of the healthcare providers.

Discussion:

This might be due to differences in “the availability and accessibility of youth-friendly health facilities, youth centers, educational status, socioeconomic status, type of residence, transportation and culture.”

Reviewer: Is this an assumption or evidence-based?

This might be due to cultural influences in the study area in which females have been still not allowed to go to health facilities for reproductive health services.

Reviewer: Grammar error. Provide evidence for this statement.

Conclusion section.

Reviewer: Authors should add on future direction of the research based on study limitations.

7. PLOS authors have the option to publish the peer review history of their article (what does this mean?). If published, this will include your full peer review and any attached files.

Reviewer #1: No

Reviewer #3: No

---

## [Author Response · Author response to Decision Letter 2]

14 Sep 2020

Dear Editors and reviewer;

Thank you very much for your critical comments and suggestions. The comments are helpful in in the shaping of the manuscript. The suggestions and comments have been closely followed and revisions have been made accordingly. The following are the questions extracted from the reviewers’ comments along with our summarized responses.

Reviewer #1. Incomplete? Extensive language editing is important not only for this section but other sections of your manuscript. ?? As you know, in urban settings these subgroup of populations are in learning institutions and all that you have said above is doing for these group (schools are settings that are best to reach adolescents). 

Author response: the abstract section has been edited and included in manuscript file. We tried to use online grammar checker.

Reviewer comment: Proper leveling of headings and subheadings of section of the manuscript may be important

Author response: heading and sub heading has been rearranged as heading one ,heading two, heading three

Reviewer comment: Sampling techniques need some revision as per comments provided .Read it what “PPS” IS really mean and revise accordingly ? Or” The calculated Sample size was proportionally allocated for selected schools” is enough for this study. In PPS, there are issues like cumulative proportion and probabilities for each corresponding cluster that is commonly suggested methods for large numbers of clusters.

Author response: The sentence “The number of respondents in the two schools (secondary and preparatory) was determined using probability proportionate to size allocation method” is replaced by “the calculated Sample size was proportionally allocated for selected schools. The sampling techniques has be revised as per the comment given.

Reviewer comment: Check epidata version 4.2.0.0

Author response: we have checked and it is epilate 4.2.0.0

Reviewer #3: The term ' sexual and reproductive' should be added to the title to reflect the content of the manuscript. I suggest 'Youth friendly sexual and reproductive health service utilization among high and preparatory school students in Debre Tabor Town, Northwest Ethiopia: a cross sectional study.

Author response: based on the comment given the title has been corrected as “Youth friendly sexual and reproductive health service utilization among high and preparatory school students in Debre Tabor Town, Northwest Ethiopia: a cross sectional study

Reviewer comment: Sample size calculation. This should be written in one paragraph. The formula is not necessary.

Author response: the formula was omitted as per the comment and summarized in one paragraph

Reviewer comment: The variables could be defined under the variables section.

Author response: really we can place under variable section, we make it under operational section to make easier for reader.

Reviewer comment: Ethical clearance was obtained from Gondar University, College of Medical and Health Sciences, Institutional Health Research Ethics Review Committee (IHRERC).Please provide the ethics approval/ registration number.

Author response: ethics approval/registration number was: Ref no.CMHS/415/2016

Reviewer comment: Was parental consent taken for those below 18 years old? If yes, this needs to be written in the manuscript.

Author response: Yes, it was included as “For children less than 18 years assent (written consent) were secured from their guardians/parents and confidentiality of the information was ensured”

Reviewer comment :294 (42.6%) of students responded that the health care facility found in their respective residences was not suitable to use the services due to the mistreating health care providers, long-distance, and unfavorable service hours.

Reviewer: Avoid begin a sentence with a number.

Author response: it is corrected as two hundred ninety four, 42.6% of students responded that the health care facility found in their respective residences was not suitable to use the services due to the mistreating by health care provider

Reviewer comment: However, 90 (13.0%) of the respondents reported that the health care providers were mistreated. Shouldn’t this be the other way round? The youths were mistreated instead of the healthcare providers.

Author response: corrected as “However, 90(13.0%) of youth were mistreated by health care providers.

Reviewer comment: This might be due to differences in “the availability and accessibility of youth-friendly health facilities, youth centers, educational status, and socioeconomic status, type of residence, transportation and culture.” Is this an assumption or evidence-based?

Author response; this is an assumption

This might be due to cultural influences in the study area in which females have been still not allowed to go to health facilities for reproductive health services. Provide evidence for this statement.

Author response: corrected as “This might be due to cultural influences in the study area in which females was less empowered. (References cited on the main document)

Reviewer comment: Authors should add future direction of the research based on study limitations.

Author response: based on this study limitation we recommend the following. So future research should be done on effect of poor utilization of reproductive health services on health outcome by including variables such as service providers’ perspective and structural barriers.

---

## [Editor Report · Decision Letter 3]

18 Sep 2020

Youth friendly sexual and reproductive health service utilization among high and preparatory school students in Debre Tabor Town, Northwest Ethiopia: a cross sectional study

PONE-D-19-33641R3

Dear Dr. Minuye,

We’re pleased to inform you that your manuscript has been judged scientifically suitable for publication and will be formally accepted for publication once it meets all outstanding technical requirements.

Kind regards,

Julie Maslowsky, PhD

Academic Editor

PLOS ONE
---

## [Editor Report · Acceptance letter]

21 Sep 2020

PONE-D-19-33641R3

Youth friendly sexual and reproductive health service utilization among high and preparatory school students in Debre Tabor Town, Northwest Ethiopia: a cross sectional study

Dear Dr. Minuye:

I'm pleased to inform you that your manuscript has been deemed suitable for publication in PLOS ONE. Congratulations! Your manuscript is now with our production department.

Kind regards,

on behalf of

Dr. Julie Maslowsky 

Academic Editor

PLOS ONE